# Lifestyle behaviors and cardiovascular risk factors in transgender versus cisgender stroke survivors

Roham Sarmadian[1], Ajith Vemuri[1], Vida Abedi[2], Ramin Zand[3]*

**1** Department of Neurology, College of Medicine, The Pennsylvania State University, Hershey, Pennsylvania, United States of America, **2** Department of Public Health Sciences, College of Medicine, The Pennsylvania State University, Hershey, Pennsylvania, United States of America, **3** Department of Neurology, College of Medicine, The Pennsylvania State University, Hershey, Pennsylvania, United States of America

* ramin.zand@gmail.com

## Abstract

Transgender individuals represent a vulnerable population with unique healthcare needs. Despite facing health disparities, there is limited research on the characteristics and stroke risk factors among transgender stroke survivors. We extracted and analyzed data from the 2020–2022 Behavioral Risk Factor Surveillance System (BRFSS) to study stroke risk factors among transgender stroke survivors and to compare them to cisgender stroke survivors. cisgender stroke survivors were paired with transgender stroke survivors using a 1:2 propensity score matching (PSM) ratio. Demographic characteristics, lifestyle factors, and vascular risk factors were assessed using univariate and multivariate logistic regression models. Out of 29,628 individuals with a history of stroke, we identified 189 transgender stroke survivors and matched them with 378 cisgender stroke survivors. Transgender individuals exhibited a higher odds of reporting a history of stroke compared to cisgenders (adjusted odds ratio[aOR]=1.692, 95% CI 1.645–1.741). Hypertension (57.8%), depression (54.3%), hypercholesterolemia (52%), and obesity (44.7%) were the most common risk factors for transgender stroke survivors. Transgender stroke survivors had higher odds of lifestyle risk factors such as e-cigarette smoking (aOR=3.651, 95%CI 1.373–9.705), binge drinking (aOR=2.519, 95%CI 1.088–5.832), alcohol consumption (aOR=2.147, 95%CI 1.049–4.395), and tobacco smoking (aOR=2.030, 95%CI 1.214–3.395). In conclusion, transgender stroke survivors may face significant challenges due to elevated modifiable risk factors, especially lifestyle-related ones. Recognizing these specific risk factors is crucial for targeted interventions for transgender stroke survivors.

## Introduction

In 2016, sexual and gender minority individuals were officially designated as a health disparity population for research by the National Institute of Health [1]. The transgender

**Data availability statement:** All data files are available from the BRFSS database.(https://www.cdc.gov/brfss/annual_data/annual_data.htm).

**Funding:** The author(s) received no specific funding for this work.

**Competing interests:** The authors have declared that no competing interests exist.

population is susceptible to health disparities and inequalities, characterized by elevated prevalence of chronic illnesses, diminished mental well-being, and access barriers to healthcare in comparison to individuals who identify as cisgender [2,3].

Stroke, a widespread cardiovascular disease, substantially impacts global health [4]. In addition to hormone therapy [5,6], several mechanisms have been proposed to explain the heightened stroke risk in transgender populations. These include a higher prevalence of traditional modifiable risk factors such as hypertension, dyslipidemia, obesity, and physical inactivity, as well as psychosocial determinants such as chronic stress, systemic discrimination, and minority stress, which can lead to atherosclerosis and thromboembolic events [7,8]. Moreover, transgender individuals often encounter fragmented or delayed care, which may contribute to suboptimal management of known vascular risk factors [9].

Most existing research on stroke risk focuses exclusively on male or female populations [10,11], largely overlooking the unique and complex factors that may influence stroke risk in transgender individuals. This oversight may result in insufficient healthcare interventions that do not take into account the particular risk profiles that transgender individuals have. While there have been limited recent studies on the risk factors associated with stroke in transgender individuals [12,13], it is also essential to assess these risk factors in post-stroke transgender individuals. Distinguishing between pre-stroke risk factors (those that predispose individuals to a first stroke) and post-stroke characteristics (ongoing or inadequately managed risk factors after stroke) is critical for the development of effective secondary prevention strategies. Understanding these distinctions helps inform whether disparities are rooted primarily in baseline exposure or in differences in post-stroke care and recovery.

This study investigates the prevalence of stroke risk factors among transgender stroke survivors and compares their characteristics and risk profiles to those of cisgender stroke survivors. Our main goal was to highlight the stroke risk factors, especially actionable risk factors, that are more prevalent among transgender stroke survivors.

## Materials and methods

### Design

Data used in this cross-sectional study were derived from the 2020–2022 Behavioral Risk Factor Surveillance System (BRFSS), an annual, nationally representative health survey conducted by the Centers for Disease Control and Prevention (CDC). As the world's largest ongoing health survey, BRFSS is administered in every U.S. state, the District of Columbia, and three U.S. territories [14]. The BRFSS survey is conducted through random-digit-dialing telephone interviews, including both landline and cell phones, and is administered year-round by state health departments with guidance from the CDC. It collects data using standardized core questions and optional modules, with results used to inform public health policies and programs. Data on various health-related behaviors, chronic diseases, and preventive health services are collected. Public access to the data used in this study is available at https://www.cdc.gov/brfss/annual_data/annual_data.htm. The data was accessed

on November 4, 2023. The study did not require ethical approval as the data is publicly available. No human subjects or animals were involved in this research, and all data used in this study conform to the principles of responsible research and data usage. Authors had no access to information that could identify individual participants. The response rates (the number of respondents who completed the survey as a proportion of all eligible and likely-eligible people) in 2020, 2021, and 2022 were 44.8%, 44.0%, and 45.1%, respectively [15–17]. Non-response bias in the BRFSS survey is addressed through weighting adjustments, imputation techniques, and continuous monitoring. Unweighted data analyses assume that each record has an equal chance of being selected and that noncoverage and nonresponse are consistent across the population. However, when these assumptions are violated and impact the results, appropriately weighting each record can help address these deviations [18]. Given this study's small final sample size, applying weights would result in significant variations in weighting factors, leading to inflated variances and unstable estimates. This could undermine the reliability of the analysis more than it would improve representativeness [19,20]. Therefore, we used unweighted data in our study.

## Study population

Adult respondents aged 18 and older are included in the BRFSS survey. In this study, only respondents who provided information regarding their gender identity were included. The two groups that the study examined were cisgender and transgender people (including male-to-female, female-to-male, and nonbinary individuals). The respondents who did not provide information about their stroke history were also excluded from the study. After that, only individuals with a history of stroke remained in the study, while those who did not have a report of a previous stroke diagnosis were excluded (Fig 1).

## Measurements

Sociodemographic data (Assigned sex at birth, age, education, income, race/ethnicity, metropolitan status, and stroke belt residency status), lifestyle risk factors (current tobacco smoking, alcohol consumption, binge drinking, e-cigarette smoking, mental health status, depression, physical activity, and body mass index), as well as vascular risk factors for stroke (diabetes, hypertension, hypercholesterolemia, and coronary artery disease) related to post-stroke respondents were extracted from the BRFSS dataset. The respondents' gender was identified based on their assigned sex at birth. As categorized by BRFSS, income levels were reported in U.S. Dollars and grouped into the following brackets: <24,999 USD/year, 25,000–49,999 USD/year, and ≥50,000 USD/year. These figures represent self-reported, unadjusted income data provided by the respondents. Metropolitan respondents were classified as those in urban areas and adjacent integrated areas, while non-metropolitan respondents were those outside metropolitan areas. Race was categorized based on the BRFSS indicators as non-Hispanic White, non-Hispanic Black, Hispanic, and other races (including American Indian, Alaskan, Asian, other minority races, and multiracial). Individuals were divided into two groups to examine the stroke belt residency status: residents of stroke belt states (including Alabama, Arkansas, Georgia, Indiana, Kentucky, Louisiana, Mississippi, North Carolina, South Carolina, Tennessee, and Virginia) and residents of other states. Information on BMI was calculated based on self-reported height and weight and categorized into underweight (BMI less than 18.5), normal weight (BMI 18.5 to less than 25.0), overweight (BMI 25.0 to less than 30.0), and obese (BMI 30.0 or higher).

The presence of chronic illnesses, including diabetes, hypertension (HTN), hypercholesterolemia, coronary artery disease (CAD), and depression, was assessed through self-report, indicating whether a healthcare provider had diagnosed these conditions in the past or not. Mental health was classified into three groups based on the length of time in the past few days that the respondents reported having suboptimal mental health: zero days (good mental health), 1–13 days (occasional poor mental health), and >14 days (Frequent mental distress). The BRFSS defines alcohol consumption as having had at least one drink of alcohol in the past 30 days. Binge drinking is defined as the consumption of 5 or more drinks on one occasion for males and 4 or more drinks for females. Low physical activity refers to a lack of exercise or

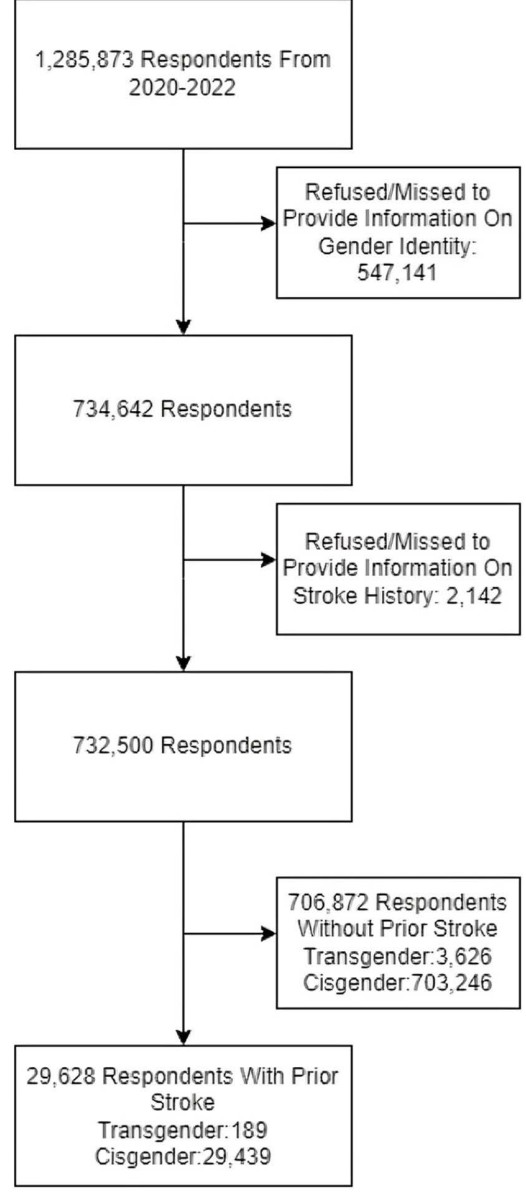

**Fig 1. Flow diagram detailing the number of participants initially enrolled, those excluded at various stages, and the final study population.**

physical activity other than regular work during the last 30 days. S1 Table presents the names and labels used in BRFSS for each of the variables utilized in the study.

## Statistical analysis

We used the 2020–2022 BRFSS annual survey data. BRFSS assigns sample weights to respondents to deal with the nonresponse and stratification-related bias.

Propensity Score Matching (PSM) was employed using the R software (version 2023.09.1) to address confounding variables and mitigate potential biases in participant selection, which could have been affected by the characteristics of stroke survivors in both the transgender and cisgender groups.

In this analysis, socio-demographic variables were considered independent variables, with gender identity (transgender or cisgender) being the dependent variable. A logistic regression model was applied to account for these factors, and a type I error rate of 0.15 was chosen to prevent the exclusion of important confounders.

The MatchIt package was used to perform propensity score matching. Propensity scores were calculated based on the significant variables (sex, age, education level, and metropolitan status), and matching was conducted. Variables such as race and income level were not included in the process due to high non-response rates and a significant reduction in sample size for matching. This methodological choice is consistent with prior BRFSS-based studies that faced similar challenges. For instance, Pierannunzi et al. describe how high rates of missing data for certain sociodemographic variables in BRFSS can necessitate excluding them from adjusted models to preserve sample size and analytic stability [21]. Moreover, due to the close similarity in stroke belt state residency status between the two groups (p = 0.936), this variable was excluded from the matching process. A total of 378 cisgender stroke survivors were paired with transgender stroke survivors using a 1:2 propensity score matching ratio through the nearest neighbor method (Fig 2).

To assess the effectiveness of PSM in reducing bias, balance measures based on prognostic scores, such as standardized mean differences (SMD), were employed. The MatchIt and cobalt packages were used to assess the balance of covariates after matching. In Table 1, the changes in SMD following PSM are observable.

IBM SPSS Statistics software version 27 was utilized for the rest of the data analysis. All categorical variables were summarized as counts and percentages. The Chi-square test was used to analyze the demographic data. Univariate and multivariate logistic regression models were used to calculate the odds ratios for risk factors. The models were adjusted for confounding variables associated with each risk factor presented in Table 3. Confounding variables in each model were selected based on clinical relevance and statistical considerations. Crude and adjusted odds ratios were calculated

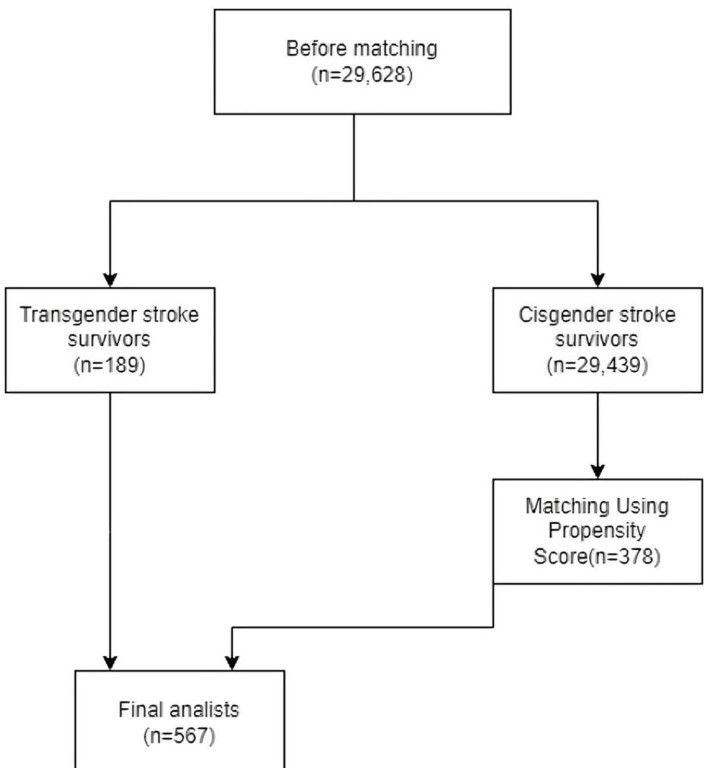

**Fig 2. Flowchart showing the number of participants before and after propensity score matching.**

**Table 1. Demographic characteristics of transgender and cisgender stroke survivors before and after propensity score matching.**

| Matched | | | | Unmatched | | | | Variable | |
|---|---|---|---|---|---|---|---|---|---|
| SMD | P-value | Cisgender stroke survivors n (%) | Transgender stroke survivors n (%) | SMD | P-value | Cisgender stroke survivors n (%) | Transgender stroke survivors n (%) | | |
| 0.016 | 0.928 | 163 (43.1) | 83 (43.9) | 0.206 | 0.006 | 15846 (54.2) | 83 (43.9) | Female | Sex (assigned at birth) |
| | | 215 (56.9) | 106 (56.1) | | | 13414 (45.8) | 106 (56.1) | Male | |
| <0.001 | >0.99 | 32 (8.5) | 16 (8.5) | 0.722 | <0.001 | 165 (0.6) | 16 (8.5) | **18-24** | Age |
| | | 52 (13.8) | 26 (13.8) | | | 511 (1.7) | 26 (13.8) | **25-34** | |
| | | 38 (10.1) | 19 (10.1) | | | 1273 (4.4) | 19 (10.1) | **35-44** | |
| | | 32 (8.5) | 16 (8.5) | | | 2839 (9.7) | 16 (8.5) | **45-54** | |
| | | 64 (16.9) | 32 (16.9) | | | 5887 (20.1) | 32 (16.9) | **55-64** | |
| | | 160 (42.3) | 80 (42.3) | | | 18585 (63.5) | 80 (42.3) | **65<** | |
| 0.020 | >0.99 | 113 (29.9) | 58 (30.7) | 0.556 | <0.001 | 3019 (10.3) | 58 (30.7) | **Did not Graduate High School** | Education Level |
| | | 117 (31.0) | 57 (30.2) | | | 9002 (30.8) | 57 (30.2) | **High school Graduate** | |
| | | 80 (21.2) | 40 (21.2) | | | 8938 (30.5) | 40 (21.2) | **Attended College** | |
| | | 66 (17.5) | 33 (17.5) | | | 8211 (28.1) | 33 (17.5) | **College Graduate** | |
| <0.001 | >0.99 | 98 (25.9) | 49 (25.9) | 0.132 | 0.094 | 9332 (31.9) | 49 (25.9) | **Non-metropolitan** | Metropolitan Status |
| | | 280 (74.1) | 140 (74.1) | | | 19928 (68.1) | 140 (74.1) | **Metropolitan** | |

to compare the odds of having different risk factors and chronic conditions between the groups. A significance level of <0.05 was considered bilaterally.

## Results

As shown in Fig 1, 1,285,783 individuals (weighted frequency of 257,079,901 individuals) participated in the survey from 2020−2022, of whom 29,628 were included in the study with a history of stroke (189 transgenders and 29,439 cisgenders). The frequency of stroke among transgender individuals was significantly greater compared to cisgenders (4.2% vs 3.4%; crude odds ratio[cOR]=1.242, 95% CI 1.235–1.249). After adjusting for demographic information, vascular and lifestyle risk factors, it was shown that the odds of reporting stroke were 69% greater (adjusted odds ratio [aOR]=1.692, 95% CI 1.645–1.741) among transgender participants.

### Demographic features

As exhibited in Table 1, in comparison to post-stroke cisgenders, the transgender stroke survivors were younger (22.3% under 35 years old vs. 2.4%; p<0.001) and had a lower level of education (30.9% without high school graduation vs. 10.4%; p<0.001). The groups had no significant difference in metropolitan status; in both groups, the number of metropolitans was higher than that of non-metropolitans. Table 2 presents the demographic variables excluded from PSM and compares them between the two groups following the matching process. After matching, it was observed that the transgender stroke survivors had a lower income (49.5% earning less than 25k vs. 34.8%; p=0.025). Transgender stroke survivors were more likely to be of Hispanic ethnicity (19.7% vs. 11%; p=0.032) or other minority races (17.1% vs. 9.0%; p=0.032). S2 Table represents the lifestyle and vascular risk factors of post-stroke transgender individuals based on their race.

### Lifestyle risk factors

Table 3 reveals that transgender stroke survivors exhibited significantly greater odds of having lifestyle risk factors in comparison to the matched cisgender stroke survivors group, except for low and high BMI, low physical activity, and

**Table 2. Comparison of other demographic characteristics in propensity score matched transgender and cisgender stroke survivors.**

| Variable | | Transgender stroke survivors n (%) | Cisgender stroke survivors n (%) | P-value |
|---|---|---|---|---|
| **Race** | **white** | 60 (51.3) | 165 (67.3) | 0.99< |
| | **Black** | 14 (12.0) | 31 (12.7) | 0.99< |
| | **Hispanic** | 23 (19.7) | 27 (11.0) | 0.034 |
| | **Other** | 20 (17.1) | 22 (9.0) | 0.034 |
| **Income Level** | **<25K USD/year** | 52 (49.5) | 71 (34.8) | 0.025 |
| | **25-50K USD/year** | 29 (27.6) | 61 (29.9) | |
| | **50K USD/year<** | 24 (22.9) | 72 (35.3) | |
| **Stroke Belt Residency Status** | **Non-stroke belt Resident** | 148 (78.3) | 298 (78.8) | 0.914 |
| | **Stroke Belt Resident** | 41 (21.7) | 80 (21.2) | |

alcohol consumption, which were not different between the two groups. Following adjustment, significantly higher odds of e-cigarette smoking (aOR=3.651, 95% CI 1.373–9.705), binge drinking (aOR=2.519, 95% CI 1.088–5.832), alcohol consumption (aOR=2.147, 95% CI 1.049–4.395), and tobacco smoking (aOR=2.030, 95% CI 1.214–3.395) were observed among transgender stroke survivors.

There were also trends toward depression and poor mental health among transgender compared to cisgender stroke survivors; however, these trends were not significant. (depression: aOR=1.873, CI=0.914–3.840, P=0.086; poor mental health: aOR=2.134, CI=0.819–5.564, P=0.121).

### Vascular risk factors

An analysis of vascular risk factors revealed that transgender individuals who had survived stroke had higher odds of having CAD (cOR=2.719, 95% CI 1.811–4.083) and diabetes (cOR=1.930, 95% CI=1.328–2.806) than cisgenders. However, upon adjustment, these variables did not differ significantly between the two groups (Table 3).

### Discussion

In this cross-sectional study, we compared post-stroke risk factors between transgender and cisgender stroke survivors. Our results indicated that transgender survivors were younger and had a lower level of education. After matching, we observed that the transgender stroke survivors had a lower income and were more likely to be of a minority race. We also found that smoking and alcohol consumption were more prevalent among transgender stroke survivors. There were also trends toward depression and poor mental health among transgender compared to cisgender stroke survivors.

Recent studies report a stroke prevalence of 1–6% among transgender individuals [3,22–24]. Among these studies, a survey by Pharr et al., using 2017–2019 BRFSS data, highlights that transgender individuals face a 2.5 times higher risk of stroke compared to cisgender individuals [3]. In our sample, 4.2% of transgender individuals reported having experienced a stroke, representing approximately 70% higher odds of having a stroke history compared to cisgender participants. Notably, while the absolute difference in stroke prevalence was relatively small, it remained statistically significant, reinforcing the clinical relevance of these disparities. Even small absolute differences can have substantial implications at the population level, particularly in vulnerable groups where multiple risk factors converge [25]. The observed association could reflect hormonal therapy used in gender transition [26–28], behavioral factors like smoking [29–31], and unique stressors such as discrimination, mental health issues, limited healthcare access, and socioeconomic disparities [29,32]. Although previous studies have associated estrogen use with a higher risk of stroke among transgender individuals, our study focuses on various post-stroke risk factors.

In our study, transgender stroke survivors had lower educational and financial resources than cisgenders. Transgender individuals often experience lower socioeconomic status compared to their cisgender counterparts, facing higher odds of

**Table 3. Comparison of stroke risk factors in propensity score matched transgender and cisgender stroke survivors.**

| Risk factor | | Transgender stroke survivors n (%) | Cisgender stroke survivors n (%) | Crude OR | 95% CI | P-value | Adjusted OR | 95% CI | P-value |
|---|---|---|---|---|---|---|---|---|---|
| Current Tobacco Smoking | | 71 (41.5) | 87 (24.2) | 2.228 | 1.511-3.285 | <0.001 | 2.030 | 1.214-3.395 | 0.007[1] |
| Alcohol Consumption | | 45 (44.0) | 95 (39.7) | 1.193 | 0.754-1.886 | 0.451 | 2.147 | 1.049-4.395 | 0.037[1] |
| Binge Drinking | | 29 (26.6) | 32 (13.5) | 2.322 | 1.320-4.086 | 0.003 | 2.519 | 1.088-5.832 | 0.031[1] |
| E-Cigarette Smoking | | 13 (21.7) | 9 (7.2) | 3.565 | 1.428-8.901 | 0.006 | 3.651 | 1.373-:9.705 | 0.009[1] |
| Depression | | 102 (54.3) | 138 (36.6) | 2.054 | 1.440-2.930 | <0.001 | 1.873 | 0.914-3.840 | 0.086[2] |
| Poor Mental Health | 0 days+ | 76 (40.6) | 177 (49.0) | – | – | – | – | – | – |
| | 1-13 days | 41 (21.9) | 82 (22.7) | 1.164 | 0.734-1.847 | 0.518 | 2.134 | 0.819-5.564 | 0.121[2] |
| | 14<days | 70 (37.4) | 102 (28.3) | 1.598 | 1.065-2.399 | 0.024 | 1.319 | 0.582-2.990 | 0.507[2] |
| Body Mass Index | Underweight | 7 (4.6) | 17 (4.9) | 1.116 | 0.422-2.949 | 0.825 | 0.428 | 0.067-2.718 | 0.368[3] |
| | Normal+ | 31 (20.4) | 84 (24.4) | – | – | – | – | – | – |
| | Overweight | 46 (30.3) | 123 (35.8) | 1.013 | 0.595-1.727 | 0.961 | 0.524 | 0.175-1.569 | 0.248[3] |
| | Obese | 68 (44.7) | 120 (34.9) | 1.535 | 0.924-2.552 | 0.098 | 1.260 | 0.475-3.345 | 0.642[3] |
| Low Physical Activity | | 73 (38.8) | 151 (40.2) | 0.946 | 0.661-1.354 | 0.761 | 0.855 | 0.502-1.457 | 0.565[1] |
| Hypertension | | 37 (57.8) | 86 (65.6) | 0.717 | 0.388-1.324 | 0.288 | 0.947 | 0.343-2.618 | 0.917[4] |
| Hypercholesterolemia | | 26 (52.0) | 61 (57.5) | 1.251 | 0.637-2.459 | 0.515 | 0.775 | 0.322-1.868 | 0.571[5] |
| Diabetes | | 74 (39.2) | 94 (25.0) | 1.930 | 1.328-2.806 | <0.001 | 1.612 | 0.696-3.729 | 0.265[6] |
| Coronary Heart Disease | | 66 (35.7) | 62 (16.9) | 2.719 | 1.811-4.083 | <0.001 | 1.605 | 0.511-5.042 | 0.417[7] |

*The reference group for the odds ratios is "Cisgender stroke survivors"

+Reference group

[1]Adjusted for race, mental health status, and income level.

[2]Adjusted for race, income level, current tobacco smoking status, alcohol consumption status, and physical activity status.

[3]Adjusted for race, income level, mental health status, current tobacco smoking status, alcohol consumption status, and physical activity status.

[4]Adjusted for race, income level, mental health status, current tobacco smoking status, alcohol consumption status, physical activity status, cholesterol status, body mass index, and blood pressure status.

[5]Adjusted for race, income level, mental health status, current tobacco smoking status, alcohol consumption status, physical activity status, cholesterol status, body mass index, and diabetes status.

[6]Adjusted for race, income level, current tobacco smoking status, alcohol consumption status, body mass index, and physical activity status.

[7]Adjusted for race, mental health status, income level, metropolitan status, current tobacco smoking status, alcohol consumption status, physical activity status, cholesterol status, body mass index, blood pressure status, and diabetes status.

unemployment, low income, and housing instability [33,34]. Furthermore, transgender individuals often face significant educational obstacles, including a high incidence of dropping out of school as a result of bullying and inadequate support [35]. This can have a substantial influence on their overall well-being.

Transgender individuals often face significant barriers to accessing appropriate healthcare, particularly when they are not socially recognized or supported [33]. A major contributing factor is the lack of cultural competence among healthcare providers [36]. According to the 2015 U.S. Transgender Survey, 23% of respondents avoided seeking medical care due to fear of mistreatment related to their transgender identity [37]. This avoidance can result in poorer health outcomes. Although the current study could not evaluate healthcare access among transgender stroke survivors, limited access or differential healthcare utilization could constitute an important explanation for some of the observed differences. Further targeted research is necessary to better understand healthcare disparities in this population.

A significant proportion of post-stroke transgender individuals belong to racial or ethnic minority groups. Overall, studies have shown that transgender adults are more likely to identify as African American, Latino, or Hispanic [38]. Racial minorities, particularly African Americans and Hispanic Americans, face a higher risk of stroke and tend to experience more

severe outcomes [39]. These disparities are exacerbated by unequal access to post-stroke care, with racial and ethnic minorities often facing significant barriers to receiving adequate treatment [40]. Therefore, it is crucial to consider racial and ethnic disparities when comparing stroke occurrences between transgender and cisgender individuals.

The proportion of individuals assigned female at birth in the cisgender group (54.1%) was similar to the proportion of individuals assigned male at birth in the transgender group (56.1%), likely due to hormone therapy. Both testosterone and estrogen can elevate cardiovascular risk, with estrogen being more strongly associated [41]. A 2021 meta-analysis found that male-to-female transgender individuals who initiated estrogen therapy had a higher risk of venous thromboembolism and ischemic stroke [42]. In the current study, data on hormone therapy use and discontinuation after stroke among transgender participants were not available; thus, we could not evaluate the role of hormone therapy in the associations. Further research is essential to better understand these associations.

Moreover, in the present study, transgender stroke survivors have shown a higher odds of tobacco smoking, alcohol use, and binge drinking, possibly due to maladaptive coping strategies [43–45]. These individuals often face significant societal challenges, which can contribute to the development of anxiety and depression [46,47]. Depression and anxiety can lead to maladaptive coping mechanisms when faced with stressful situations such as stroke [48,49]. Because of the cross-sectional design, it is not possible to determine whether substance use preceded stroke, resulted from stroke, or both; reverse causation and bidirectional relationships are plausible. Longitudinal investigations are required to better understand temporal relationships.

Additionally, our results indicate a significant disparity in e-cigarette use, with transgender stroke survivors being nearly four times more likely to use e-cigarettes compared to their cisgender counterparts. Although often promoted as a safer alternative or smoking cessation aid, growing evidence suggests that e-cigarette use is associated with increased cardiovascular and cerebrovascular risks, particularly due to nicotine exposure [50,51]. Several studies reported a significant association between e-cigarette use and increased stroke risk [52,53]. Moreover, dual use of traditional cigarettes and e-cigarettes was found to further elevate cardiovascular risk [54]. These findings underscore the need to address e-cigarette use in stroke prevention strategies, particularly among vulnerable populations such as transgender individuals.

Although depression and poor mental health were more common among transgender stroke survivors, these differences were not statistically significant. Nonetheless, mental health remains an important factor in stroke risk. Transgender individuals experience increased psychological distress, which may contribute to stroke through indirect pathways such as substance use and care avoidance [55–57]. Future longitudinal and qualitative studies are needed to clarify these associations and improve prevention strategies.

Our findings also show that the increased odds of reporting stroke in transgender individuals persist after adjusting for risk factors. This suggests that additional factors, including hormonal influence and healthcare disparities, may contribute to the observed disparities. It is noteworthy that for some post-stroke risk factors analyzed, substantial differences were observed between crude and adjusted ORs, highlighting the presence of confounding variables and the importance of multivariable analysis in this context.

This study is the first to specifically investigate post-stroke risk factors among transgender individuals. However, several limitations should be noted. The relatively small sample size of transgender stroke survivors (n = 189) may limit the generalizability of our findings and highlights the need for larger, more representative studies to confirm these results. The study lacks data on hormone and illicit drug use and comparisons of healthcare service utilization, as these variables were not included in the BRFSS survey. Factors such as depression, e-cigarette use, high cholesterol, and hypertension were assessed only in the 2021 BRFSS survey, resulting in a smaller sample size for these variables. Moreover, the reliance on self-reported data without clinical or objective validation introduces potential for misclassification bias, not only for stroke diagnosis but also for other variables, particularly stigmatized conditions such as mental health disorders, substance use, and gender identity-related factors. Additionally, BRFSS does not distinguish between ischemic and hemorrhagic stroke. Future studies should validate findings using clinically confirmed data to improve reliability. Furthermore, the reliance on

telephone interviews may have introduced bias due to stigma, potentially affecting the representation of transgender individuals and associated risk factors. Survivor bias may also be present, as individuals with the most severe strokes may not have survived to be included in a telephone survey, possibly underestimating true disparities. Because BRFSS data are collected via telephone interviews, the temporal sequence of stroke and associated risk factors remains unclear. It is not possible to determine whether these factors developed prior to stroke onset or emerged afterward. Similarly, the timing of gender identity development relative to stroke and risk factor accumulation cannot be established from cross-sectional data. Finally, it is important to note that given the multiple comparisons performed, there is an increased risk of Type I error; therefore, p-values should be interpreted with caution.

Future studies are needed to explore further the underlying causes of increased stroke incidence among transgender individuals. Important areas include the impact of gender-affirming hormone therapy, barriers to healthcare access, and the influence of psychosocial stressors such as stigma and discrimination. Longitudinal research is particularly important to better understand the progression of risk factors. Additionally, examining the role of race, socioeconomic status, and behaviors such as tobacco, alcohol, and e-cigarette use may help identify targeted strategies to reduce disparities and improve care in this underserved population.

## Conclusions

In conclusion, this study highlights significant disparities in post-stroke risk factors and characteristics among transgender individuals compared to their cisgender counterparts. Transgender stroke survivors were found to have lower educational attainment and income and were more likely to belong to racial or ethnic minority groups. Additionally, they exhibited higher odds of tobacco smoking, alcohol consumption, and e-cigarette use, suggesting potential behavioral and psychosocial contributors to their increased stroke risk. Our study underscores the need for a comprehensive approach that considers lifestyle, socioeconomic factors, and mental health in post-stroke care for transgender individuals. Future efforts should prioritize the development of risk reduction strategies specifically tailored to transgender stroke survivors. These may include routine screening for modifiable risk factors, culturally sensitive cessation programs for tobacco and alcohol use, improved access to mental health services, and integration of gender-affirming care into rehabilitation pathways.

## Supporting information

**S1 Table. Names and labels used for the variables within BRFSS.**
(DOCX)

**S2 Table. Comparison of risk factors between white transgender stroke survivors and other racial groups.**
(DOCX)

## Acknowledgments

We would like to thank Dr. Amir Hamta, Assistant Professor of Biostatistics at Arak University of Medical Sciences, for providing consultation on statistical analyses.

## Author contributions

**Conceptualization:** Roham Sarmadian.

**Data curation:** Roham Sarmadian, Ajith Vemuri, Vida Abedi, Ramin Zand.

**Formal analysis:** Roham Sarmadian.

**Investigation:** Roham Sarmadian.

**Methodology:** Roham Sarmadian, Ramin Zand.

**Project administration:** Ramin Zand.

**Software:** Roham Sarmadian.

**Supervision:** Ajith Vemuri, Ramin Zand.

**Validation:** Roham Sarmadian, Ajith Vemuri, Vida Abedi, Ramin Zand.

**Visualization:** Roham Sarmadian.

**Writing – original draft:** Roham Sarmadian.

**Writing – review & editing:** Roham Sarmadian, Ajith Vemuri, Vida Abedi, Ramin Zand.

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
