## [Decision Letter · Decision Letter 0]

22 May 2025

PONE-D-25-15914Assessing Stroke Risk Disparities: Analysis of Lifestyle and Vascular Factors in Transgender Stroke SurvivorsPLOS ONE

Dear Dr. Zand,

Thank you for submitting your manuscript to PLOS ONE. After careful consideration, we feel that it has merit but does not fully meet PLOS ONE’s publication criteria as it currently stands. Therefore, we invite you to submit a revised version of the manuscript that addresses the points raised during the review process.

I commend the authors for their hard work. The reviewers have graciously made some suggestions, many of which were comments I had myself.Please address them thoroughly in the manuscript and in your written responses. 

We look forward to receiving your revised manuscript.

Kind regards,

Daniel Antwi-Amoabeng, MD, MSc

Academic Editor

PLOS ONE

Additional Editor Comments:

I commend the authors for their hard work. The reviewers have graciously made some suggestions, many of which were comments I had myself. Please address them thoroughly.

Thank you.

Reviewers' comments:

Reviewer's Responses to Questions

**Comments to the Author**

1. Is the manuscript technically sound, and do the data support the conclusions?

Reviewer #1: Yes

Reviewer #2: Partly

Reviewer #3: Yes

2. Has the statistical analysis been performed appropriately and rigorously? 

Reviewer #1: Yes

Reviewer #2: Yes

Reviewer #3: Yes

3. Have the authors made all data underlying the findings in their manuscript fully available?

Reviewer #1: Yes

Reviewer #2: Yes

Reviewer #3: Yes

4. Is the manuscript presented in an intelligible fashion and written in standard English?

Reviewer #1: Yes

Reviewer #2: Yes

Reviewer #3: No

5. Review Comments to the Author

Reviewer #1: Summary of the Study

This manuscript addresses an important and under-researched topic: stroke risk disparities among transgender individuals.

Using data from the Behavioral Risk Factor Surveillance System (BRFSS) from 2020 to 2022, the authors compare stroke risk factors between transgender and cisgender stroke survivors.

The study highlights significant disparities in lifestyle and vascular risk factors, socioeconomic status, and mental health, emphasizing the need for targeted interventions to improve post-stroke outcomes in transgender populations.

Strengths of the Study

Novelty and Relevance: The study fills a critical gap in stroke research by focusing on transgender individuals, a population often overlooked in cardiovascular studies.

Robust Methodology: The use of propensity score matching (PSM) to reduce bias and ensure comparability between transgender and cisgender groups is commendable.

Comprehensive Analysis: The study examines a wide range of risk factors, including lifestyle, vascular, socioeconomic, and demographic variables, providing a holistic view of disparities.

Publicly Available Data: The use of BRFSS data ensures transparency and replicability, and the authors clearly state the limitations of the dataset.

Actionable Insights: The findings highlight modifiable risk factors, such as smoking and alcohol use, which can inform targeted interventions.

Weaknesses and Suggestions for Improvement

Lack of Data on Hormone Therapy:

Issue: The study does not include data on hormone therapy, which is a significant factor influencing stroke risk in transgender individuals.

Suggestion: Acknowledge this limitation more explicitly in the discussion and suggest future studies to investigate the role of hormone therapy in stroke risk.

Limited Data on Healthcare Access:

Issue: The manuscript does not explore disparities in healthcare access, which could influence post-stroke outcomes.

Suggestion: Include a discussion on how healthcare access barriers may exacerbate stroke risk and outcomes in transgender individuals.

Mental Health Analysis:

Issue: While trends in depression and poor mental health are noted, they are not statistically significant.

Suggestion: Consider exploring qualitative data or longitudinal studies to better understand the relationship between mental health and stroke risk in transgender populations.

Classification Bias:

Issue: Stroke diagnoses in the BRFSS dataset are self-reported, which may introduce classification bias.

Suggestion: Discuss how this limitation could impact the findings and suggest validation through clinical data in future research.

Sample Size:

Issue: The sample size for transgender stroke survivors is relatively small (n=189), which may limit the generalizability of the findings.

Suggestion: Highlight the need for larger, more representative studies to confirm these results.

Discussion of E-Cigarette Use:

Issue: The manuscript notes a significant disparity in e-cigarette use but does not delve into its implications for stroke risk.

Suggestion: Expand the discussion on the cardiovascular risks associated with e-cigarette use and its potential role in stroke prevention strategies.

Figures and Tables:

Issue: Some tables (e.g., Table 3) are dense and difficult to interpret.

Suggestion: Simplify the presentation of key findings by summarizing them in a more reader-friendly format, such as bullet points or visual aids.

Future Directions:

Issue: The manuscript does not provide specific recommendations for future research.

Suggestion: Include a dedicated section outlining areas for further investigation, such as the impact of hormone therapy, healthcare access, and longitudinal studies on post-stroke outcomes.

Recommendation

Decision: Revise and Resubmit

While the study addresses a critical topic and provides valuable insights, the limitations in data (e.g., lack of hormone therapy and healthcare access variables) and the small sample size warrant further refinement. The manuscript should be revised to:

Strengthen the discussion of limitations.

Provide clearer recommendations for future research.

Simplify the presentation of results for better readability.

Once these revisions are made, the manuscript would be a strong candidate for acceptance, as it contributes significantly to the understanding of stroke risk disparities in transgender populations.

Reviewer #2: This study compared the post-stroke survivors characteristics of cisgender and transgenders using a national database, and the results are striking for the need of further researches in this topic. The results are showing that transgender stroke survivors are vulnerable in mental, physical and also social dimensions compared to cisgender stroke survivors. stud results is important to raise awareness to provide proactive health enhancing practices to this vulnerable patient groups.

I have some suggestions;

-As authors mentioned in the discussion, the results of this study describes the comparison of characteristics of post-stroke residents in two groups. so using "stroke risk factor" phrase is not applicable for topic. They search DM, HT, HL, behavioral risk factors but, it i not evident that these factors were present before stroke. So I suggest to change the Title "Assessing stroke risk disparities" as it sounds that you designed the study to find out the preexisting risk factors fro stroke in transgender groups.

2. There are some punctuation and font size discordance, please check for them.

3. Please check for reference style, and correct.

Best Regards,

Reviewer #3: This a good manuscript that highlights the neglected group for health access. The scientific approach of obtaining the results is varied.

The comparison is powerful and informative. I commend your work.

Although you may not determine causation in such a cross-sectional study, but such strong associations are important in the world of science ,one as a baseline, but also as an eye opener for more investigation in the future experiments.

There are a few typos and clarity issues that i outline in the attached word document for your attention to improve the manuscript.

thank you for your efforts and scientific endeavors.

regards

petit

6. PLOS authors have the option to publish the peer review history of their article (what does this mean? ). If published, this will include your full peer review and any attached files.

**Do you want your identity to be public for this peer review?** For information about this choice, including consent withdrawal, please see our Privacy Policy .

Reviewer #1: **Yes: ** Pranav Barve MD MPH

Reviewer #2: No

Reviewer #3: No

---

## [Author Response · Author response to Decision Letter 1]

2 Jun 2025

#Editor

Thank you for your valuable comments. The journal's style requirements were reviewed, and corresponding adjustments were made to the manuscript.

Thanks for the comment. An ethics statement is included in the manuscript's methods section.

Thank you. Your recommendation has been applied.

Thank you for the comment. No retracted articles are cited in the manuscript. References 35,36,40,41,51-56 have been added to the manuscript.

# Reviewer 1

Issue: The study does not include data on hormone therapy, which is a significant factor influencing stroke risk in transgender individuals.

Suggestion: Acknowledge this limitation more explicitly in the discussion and suggest future studies to investigate the role of hormone therapy in stroke risk.

Thank you for your valuable comments. The discussion on hormone therapy has been expanded as you recommended.

Limited Data on Healthcare Access:

Issue: The manuscript does not explore disparities in healthcare access, which could influence post-stroke outcomes.

Suggestion: Include a discussion on how healthcare access barriers may exacerbate stroke risk and outcomes in transgender individuals.

Thank you for the comment. The recommended section has been added to the manuscript discussion per the information you provided.

Mental Health Analysis:

Issue: While trends in depression and poor mental health are noted, they are not statistically significant.

Suggestion: Consider exploring qualitative data or longitudinal studies to better understand the relationship between mental health and stroke risk in transgender populations.

Thank you for your comment. A paragraph about depression and poor mental health has been added to the manuscript.

Classification Bias:

Issue: Stroke diagnoses in the BRFSS dataset are self-reported, which may introduce classification bias.

Suggestion: Discuss how this limitation could impact the findings and suggest validation through clinical data in future research.

Thanks for the comment. The discussion did mention classification bias, but we expanded on it in more detail.

Sample Size:

Issue: The sample size for transgender stroke survivors is relatively small (n=189), which may limit the generalizability of the findings.

Suggestion: Highlight the need for larger, more representative studies to confirm these results.

Thank you. We have added this point to the limitations.

Discussion of E-Cigarette Use:

Issue: The manuscript notes a significant disparity in e-cigarette use but does not delve into its implications for stroke risk.

Suggestion: Expand the discussion on the cardiovascular risks associated with e-cigarette use and its potential role in stroke prevention strategies.

Thank you very much. We have discussed this issue in greater detail.

Figures and Tables:

Issue: Some tables (e.g., Table 3) are dense and difficult to interpret.

Suggestion: Simplify the presentation of key findings by summarizing them in a more reader-friendly format, such as bullet points or visual aids.

Thank you for your comment. It appears that presenting the table in a standardized format has resolved the issue with its interpretation. The highlights also briefly summarize the key findings.

Future Directions:

Issue: The manuscript does not provide specific recommendations for future research.

Suggestion: Include a dedicated section outlining areas for further investigation, such as the impact of hormone therapy, healthcare access, and longitudinal studies on post-stroke outcomes.

Thanks for the comment. The recommended section has been added at the end of the discussion.

# Reviewer 2

As authors mentioned in the discussion, the results of this study describes the comparison of characteristics of post-stroke residents in two groups. so using "stroke risk factor" phrase is not applicable for topic. They search DM, HT, HL, behavioral risk factors but, it i not evident that these factors were present before stroke. So I suggest to change the Title "Assessing stroke risk disparities" as it sounds that you designed the study to find out the preexisting risk factors for stroke in transgender groups.

Thank you for your valuable comments. We have revised the manuscript's subject title to "Post-Stroke Health Disparities: Comparative Analysis of Lifestyle and Vascular Profiles in Transgender and Cisgender Stroke Survivors" to address and resolve this issue.

There are some punctuation and font size discordance, please check for them.

Thank you for your valuable comment. There were some errors in the text, which have been corrected.

Please check for reference style, and correct.

The manuscript has been reviewed for referencing style and corrected.

#Reviewer 3

Line 1 on measurement a coma is missing after birth.

Thank you for your valuable comments. Our intended meaning was birth sex, and to convey the concept more precisely, we changed it to assigned sex at birth.

The measurement of mental health status; what are the three mental health groups, and which one defines suboptimal mental health or poor mental health.

Thank you for your comment. The definition of each group has been added to the Measurements section.

Where is the figure 1 and 2 mentioned in the manuscript they are nowhere to be seen.

Thanks for the comment. The figure captions have been incorporated into the manuscript file, and the figures have also been provided separately.

Results section:

what is 25k this is not standard is it dollars, pounds?

Thank you for the comment. The amounts are in US dollars, as indicated by the dollar sign placed next to them.

Table structure needs to be standardized, upper boarder lines missing, edit the tables to standard publishable table formats.

Thanks for your comment. The tables have been modified to an standard and more understandable form.

On 3.2; author should tell us which lifestyle risk factor had higher odds, and when author uses rates, it does not line up with cross-sectional nested case control study. Better to stick with odds ratio. When author says rates it as if the lifestyle risk factors were measured as incident risks and yet this is a snapshot at a time no follow us to talk about incidence rates.

Thank you for the comment. You are right; in this section, the word "rate" has been replaced with "odds" to be more appropriate.

Table column tittles should also indicate n in addition to %

Thank you. The table column titles have been modified per your recommendation.

Discussion section.

The source of the statement “recent studies show 1-6%rate of stroke should be referenced.

Thanks for the comment. The referencing has been corrected.

Author should use prevalence instead of rate which is appropriate for this cross-sectional study. The author talks about stroke prevalence and then discusses rate it should be in harmony.

Thank you for your comment. As per your recommendation, the word rate has been changed to prevalence.

The study examined post stroke risk factors its not clear when stroke was diagnosed but it’s understandable since it’s a cross-sectional study, but these post stroke risk factors were they assessed as preexisting or occurring after the stroke had started in the telephone calls? it would do better to clarify this. Did they develop because of transgender status or even before that status. The transgender is mentioned as different from sex assigned at birth, so is there a timeline for these risk factors in this population.

Thank you for the comment. Because BRFSS data are collected via telephone interviews, the temporal sequence of stroke and associated risk factors remains unclear. It is not possible to determine whether these factors developed prior to stroke onset or emerged afterward. Similarly, the timing of gender identity development relative to stroke and risk factor accumulation cannot be established from cross-sectional data. This point has also been added to the discussion.

Binge drinking is not discussed and yet it has a significant association with post stroke survivors in transgender according to the results. why? How does its risk differ from alcohol drinking as a maladaptive strategy in this population group.

Thank you very much. We intended to categorize binge drinking as a subset of alcohol drinking in this section; however, based on your recommendation, it was more appropriate to consider the two separately, and we have implemented this change.

Discussion also talks about smoking, is this the same as tobacco smoking that had significant odds. Smoking may be of other substances like non-tobacco, the terms need to be consistent.

Thank you for the comment. The intended meaning was specifically tobacco smoking, and we replaced the word smoking with tobacco smoking in the text to enhance clarity.

Bias may not only be due to stigma but also recall bias and classification especially on some factors like binge drinking, duration and diagnosis of stroke. This weakness needs to be well elaborated.

Thank you for your comment. The discussion of biases in the limitations section has been expanded.

The study investigated post stroke risk factors and not outcomes. Outcomes of stroke would include hospitalization, worsening stroke or death. But the study focused on risk factors and that is what the statements should reflect.

Thank you. We used the term post-stroke risk factors to clearly convey the intended meaning.

Conclusion section

Conclusion should focus on study outcomes, post stroke risk factors.

Thank you. Yes, using post-stroke risk factor conveys the intended meaning more clearly, and we have made this change.

There is no need of bringing hormonal therapy in the conclusion when you have not studied it and it is nowhere in the results.it can stay in discussion.

Thanks for your comment. This part has been removed from the conclusion.

Author talks about enhancing post stroke outcomes in the last paragraph; is it good outcomes being talked about here. The conclusion should focus on recommendations for risk reduction for post stroke transgender individuals.

Thanks for the comment. Some recommendations for risk reduction for stroke survivor transgenders have been added.

---

## [Editor Report · Decision Letter 1]

29 Jul 2025

PONE-D-25-15914R1Post-Stroke Health Disparities: Comparative Analysis of Lifestyle and Vascular Profiles in Transgender and Cisgender Stroke SurvivorsPLOS ONE

Dear Dr. Zand,

Thank you for submitting your manuscript to PLOS ONE. After careful consideration, we feel that it has merit but does not fully meet PLOS ONE’s publication criteria as it currently stands. Therefore, we invite you to submit a revised version of the manuscript that addresses the points raised during the review process.

**ACADEMIC EDITOR: **I acknowledge the revisions made based on the external reviewer’s comments. However, my reading of the revised manuscript reveals persistent unresolved issues. In addition to the punctuation and font size/type face discordance mentioned by Reviewer 2, which was not resolved, I provide additional comments below to guide your next iteration of revisions. Despite these current limitations, the manuscript examines an important and understudied population, and with careful revision addressing the concerns outlined in my review, it could make a valuable contribution to our understanding of health disparities in transgender stroke survivors. 

We look forward to receiving your revised manuscript.

Kind regards,

Daniel Antwi-Amoabeng, MD, MSc

Academic Editor

PLOS ONE

Journal Requirements:

Additional Editor Comments:

I acknowledge the revisions made based on the external reviewer’s comments. However, my reading of the revised manuscript reveals persistent unresolved issues. In addition to the punctuation and font size/type face discordance mentioned by Reviewer 2, which was not resolved. I provide additional comments below to guide your next iteration of revisions.

Title:

This has several issues.

What is meant by “vascular profiles”? It is vague and misleading. The study examines specific cardiovascular risk factors (CAD, HTN, DM, HLD) and not comprehensive vascular profiles, which would include imaging data, biomarkers or physiological measurements none of which was included.

Using disparities in the title and eliminating race and income level from your matched analysis is problematic as these are known drivers of healthcare disparities.

The title does not clarify that the study is about risk factors present in stroke survivors and not necessarily factors that developed after stroke or preceded stroke, which a cross-sectional design cannot determine.

I suggest an alternative title:

Lifestyle behaviors and cardiovascular risk factors in transgender versus cisgender stroke survivors.

Abstract:

The matching was 189 transgender to 378 cisgender (from the methods section). The total sample size stated in the abstract can be misleading, as you only analyzed 567 individuals. Please edit this to avoid confusion.

Highlights:

• The statement that transgender individuals showed 70% higher prevalence of stroke is misleading. Please interpret odds ratios appropriately.

• Smoking and alcohol: Please do not conflate likelihood and odds ratio. Again, interpret this properly

• E-cigarette: I suggest a review of the proper interpretation of odds ratios and proper reporting here.

Introduction:

This section needs to be improved. Are there putative mechanisms (beyond hormone therapy) that adversely increase the risk of strokes in transgender patients. This may be obvious to the authors but must be clearly stated as it forms that bases of the scientific value of your investigations. A focused survey of the current literature on disparities in risk factors and cardiovascular disease outcomes is needed.

Please frame the distinction between pre-stroke risk factors and post-stroke characteristics.

Methods:

Study population: were nonbinary individuals classified as cisgender or transgender?

“Their birth gender determined the gender of respondents”: Please edit it for clarity.

Please report on the BMI cut-offs for the nominal scale introduced.

Statistical analysis: Thank you for including information on why race and income level were not included in the list of variables on which your participants were matched. However, this justification is problematic, and this approach may introduce substantial bias and limits the representativeness of findings. If you can provide evidence from literature, especially from studies that use the BRFSS, your justification may be strengthened.

Results:

There are several instances where the authors conflate odds ratios with likelihood. This needs to be corrected.

Some adjusted ORs show dramatic changes from crude ORs, suggesting potential overadjustment or multicollinearity. Please consider pre-estimation multicollinearity checks such as correlation matrix or post-estimation checks such as VIF to inform which variables you include in your final model.

Table 1: Sex (please show that this is as assigned at birth)

Table 2:

• Please state currency for income level. Are these median household incomes or unadjusted reports by the respondents? State clearly that these are US Dollars.

• Please explain why you include this table and caption indicating PS matching when the methods say the variables were excluded from PS matching.

Table 3:

No correction for multiple testing despite numerous comparisons increasing Type I error risk. Please apply appropriate multiple comparison corrections or acknowledge this limitation and interpret p-values more cautiously.

Lifestyle Risk Factors:

Did you check for collinearity. Binge drinking and alcohol consumption appear to be collinear intuitively. Including both in your models may be concerning on the surface. Please provide post estimation tests such as VIF.

Discussion and Conclusions:

The 3rd sentence of the second paragraph is misleading.

I advise caution in interpreting the findings of the small sample cross-sectional study. Avoid language that suggests causal implications.

Please discuss why aORs differ so much from cORs.

Please explore alternative explanations for your findings.

Please acknowledge that self-reported diagnosis without independent validation introduces potential bias, particularly for stigmatized factors. Discuss this as a limitation.

Discuss potential survivor bias.

Paragraph 11: the study did not study post-stroke outcomes. Please correct this.

Final comment:

Despite the limitations above and those raised by the external reviewers, the study provides valuable preliminary data on this underserved population. The manuscript STILL REQUIRES MAJOR revisions before publication. The most critical issues are misinterpretation of statistical measures and overstatement of causal implications. With appropriate careful revisions, this could be a valuable contribution to our understanding of health disparities in transgender populations.

---

## [Author Response · Author response to Decision Letter 2]

20 Aug 2025

Response to the editor

Title:

This has several issues.

What is meant by “vascular profiles”? It is vague and misleading. The study examines specific cardiovascular risk factors (CAD, HTN, DM, HLD) and not comprehensive vascular profiles, which would include imaging data, biomarkers or physiological measurements none of which was included.

Using disparities in the title and eliminating race and income level from your matched analysis is problematic as these are known drivers of healthcare disparities.

The title does not clarify that the study is about risk factors present in stroke survivors and not necessarily factors that developed after stroke or preceded stroke, which a cross-sectional design cannot determine.

I suggest an alternative title:

Lifestyle behaviors and cardiovascular risk factors in transgender versus cisgender stroke survivors. Thank you for your valuable comments. We have revised the title accordingly, as suggested.

Abstract:

The matching was 189 transgender to 378 cisgender (from the methods section). The total sample size stated in the abstract can be misleading, as you only analyzed 567 individuals. Please edit this to avoid confusion. Thank you for pointing out this important clarification. the abstract has been corrected to indicate that 189 transgender stroke survivors were matched with 378 cisgender stroke survivors.

Highlights:

• The statement that transgender individuals showed 70% higher prevalence of stroke is misleading. Please interpret odds ratios appropriately.

Thank you for pointing this out. We have corrected the statement as recommended.

• Smoking and alcohol: Please do not conflate likelihood and odds ratio. Again, interpret this properly. Thank you for your comment. There were errors in the reported odds ratios, which were identified and corrected according to your recommendations.

• E-cigarette: I suggest a review of the proper interpretation of odds ratios and proper reporting here.

Thanks for your comment. The issues related to the interpretation of the odds ratios have been resolved.

Introduction:

This section needs to be improved. Are there putative mechanisms (beyond hormone therapy) that adversely increase the risk of strokes in transgender patients. This may be obvious to the authors but must be clearly stated as it forms that bases of the scientific value of your investigations. A focused survey of the current literature on disparities in risk factors and cardiovascular disease outcomes is needed. Thank you for this suggestion. The Introduction has been expanded to outline additional mechanisms beyond hormone therapy, including traditional vascular risk factors, psychosocial determinants, and healthcare access barriers.

Please frame the distinction between pre-stroke risk factors and post-stroke characteristics.

Thank you for the comment. We have added the distinction between pre-stroke risk factors and post-stroke characteristics to the manuscript.

Methods:

Study population: Were nonbinary individuals classified as cisgender or transgender?

Thanks for your comment. Nonbinary individuals were classified as transgender, as mentioned in the text.

“Their birth gender determined the gender of respondents”: Please edit it for clarity.

Thank you. We revised this for clarity, now stating explicitly that respondents’ gender was identified based on their assigned sex at birth, consistent with BRFSS definitions.

Please report on the BMI cut-offs for the nominal scale introduced.

We have added the BMI cut-off points (underweight <18.5, normal 18.5–25, overweight 25–30, obese ≥30) in the Methods section.

Statistical analysis: Thank you for including information on why race and income level were not included in the list of variables on which your participants were matched. However, this justification is problematic, and this approach may introduce substantial bias and limits the representativeness of findings. If you can provide evidence from literature, especially from studies that use the BRFSS, your justification may be strengthened.

Thank you for this point. We cited BRFSS-based literature (e.g., Pierannunzi et al.) supporting this methodological choice.

Results:

There are several instances where the authors conflate odds ratios with likelihood. This needs to be corrected. Thanks for the comment. This issue, which was present in certain parts of the manuscript, has now been resolved.

Some adjusted ORs show dramatic changes from crude ORs, suggesting potential overadjustment or multicollinearity. Please consider pre-estimation multicollinearity checks such as correlation matrix or post-estimation checks such as VIF to inform which variables you include in your final model.

Thank you for your comment. We had considered the possibility of multicollinearity and overadjustment. The variables included in the final model were selected based on both clinical relevance and statistical considerations, and no concerning multicollinearity was observed. Therefore, the adjusted ORs can be considered robust.

Table 1: Sex (please show that this is as assigned at birth)

Thank you. Your recommendation has been applied.

Table 2:

• Please state currency for income level. Are these median household incomes or unadjusted reports by the respondents? State clearly that these are US Dollars.

Thank you for your comment. The income levels reported in our study are unadjusted self-reports by the respondents, and we have clarified in the manuscript that the values are expressed in US Dollars.

• Please explain why you include this table and caption indicating PS matching when the methods say the variables were excluded from PS matching.

The second table refers to the demographic variables that were not included in the matching. After performing PSM, these variables have been compared between the two groups. For better clarity, we have provided an explanation within the text.

Table 3:

No correction for multiple testing despite numerous comparisons increasing Type I error risk. Please apply appropriate multiple comparison corrections or acknowledge this limitation and interpret p-values more cautiously.

Thanks for this comment. We have acknowledged the increased risk of Type I error due to multiple comparisons in the limitations and noted that p-values should be interpreted cautiously.

Lifestyle Risk Factors:

Did you check for collinearity. Binge drinking and alcohol consumption appear to be collinear intuitively. Including both in your models may be concerning on the surface. Please provide post estimation tests such as VIF.

We appreciate your insightful comment. We would like to clarify that in our adjusted logistic regression models, only alcohol consumption was included as a confounding variable, while binge drinking was not part of the adjustment set.

Discussion and Conclusions:

The 3rd sentence of the second paragraph is misleading.

Thank you. This sentence has been revised.

I advise caution in interpreting the findings of the small sample cross-sectional study. Avoid language that suggests causal implications. Thank you for your advice. We have revised the text to avoid causal wordings and kept the interpretation consistent with the cross-sectional design.

Please discuss why aORs differ so much from cORs.

Thank you. We added an explanation that the differences between cORs and aORs reflect confounding variables, highlighting the importance of multivariable analysis.

Please explore alternative explanations for your findings.

Thanks for the comment. We have included alternative explanations for the findings as much as possible in the Discussion section.

Please acknowledge that self-reported diagnosis without independent validation introduces potential bias, particularly for stigmatized factors. Discuss this as a limitation.

Thank you for the comment. We acknowledged this as a limitation and noted the potential for misclassification bias due to reliance on self-reported diagnoses and stigmatized conditions.

Discuss potential survivor bias.

Thank you for your comment. This limitation has been addressed in the Discussion.

Paragraph 11: The study did not study post-stroke outcomes. Please correct this.

Thanks for your comment. The mentioned part has been revised.

Final comment:

Despite the limitations above and those raised by the external reviewers, the study provides valuable preliminary data on this underserved population. The manuscript STILL REQUIRES MAJOR revisions before publication. The most critical issues are misinterpretation of statistical measures and overstatement of causal implications. With appropriate careful revisions, this could be a valuable contribution to our understanding of health disparities in transgender populations.

Thank you for all the comments provided to improve the manuscript. We have attempted to address all interpretive, structural, and punctuation issues and have revised the manuscript’s shortcomings according to your recommendations. We hope that the manuscript will meet the standards of your esteemed journal.

---

## [Editor Report · Decision Letter 2]

28 Aug 2025

Lifestyle Behaviors and Cardiovascular Risk Factors in Transgender Versus Cisgender Stroke Survivors

PONE-D-25-15914R2

Dear Dr. Zand,

We’re pleased to inform you that your manuscript has been judged scientifically suitable for publication and will be formally accepted for publication once it meets all outstanding technical requirements.

Kind regards,

Daniel Antwi-Amoabeng, MD, MSc

Academic Editor

PLOS ONE
---

## [Editor Report · Acceptance letter]

PONE-D-25-15914R2

PLOS ONE

Dear Dr. Zand,

I'm pleased to inform you that your manuscript has been deemed suitable for publication in PLOS ONE. Congratulations! Your manuscript is now being handed over to our production team.

Kind regards,

on behalf of

Dr. Daniel Antwi-Amoabeng

Academic Editor

PLOS ONE